# A Multiscale Gradient Fusion Method for Color Image Edge Detection Using CBM3D Filtering

**DOI:** 10.3390/s25072031

**Published:** 2025-03-24

**Authors:** Zhunruo Feng, Ruomeng Shi, Yuhan Jiang, Yiming Han, Zeyang Ma, Yuheng Ren

**Affiliations:** 1School of Electronics and Information, Xi’an Polytechnic University, Xi’an 710048, China; lihaifzr@163.com (Z.F.); h3130822503@163.com (Y.H.); mzy3229@163.com (Z.M.); 2School of International Business School Suzhou, Xi’an Jiaotong Liverpool University, Suzhou 215123, China; ruomengs@outlook.com; 3School of the Arts, Universitiy Sains Malaysia, Penang 11700, Malaysia; 18182052185@163.com; 4School of Digital Industry, Jimei University, Xiamen 361021, China

**Keywords:** image edge detection, CBM3D filter, anisotropic Gaussian derivative, XYZ color model

## Abstract

In this paper, we present a novel color edge detection method that integrates collaborative filtering with multiscale gradient fusion. The Block-Matching and 3D (BM3D) filter is utilized to enhance sparse representations in the transform domain, effectively reducing noise. The multiscale gradient fusion technique compensates for the loss of detail in single-scale edge detection, thereby improving both edge resolution and overall quality. RGB images from the dataset are converted into the XYZ color space through mathematical transformations. The Colored Block-Matching and 3D (CBM3D) filter is applied to the sparse images to reduce noise. Next, the vector gradients of the color image and anisotropic Gaussian directional derivatives for two scale parameters are computed. These are then averaged pixel-by-pixel to generate a refined edge strength map. To enhance the edge features, the image undergoes normalization and non-maximum suppression. This is followed by edge contour extraction using double-thresholding and a novel morphological refinement technique. Experimental results on the edge detection dataset demonstrate that the proposed method offers robust noise resistance and superior edge quality, outperforming traditional methods such as Color Sobel, Color Canny, SE, and Color AGDD, as evidenced by performance metrics including the PR curve, AUC, PSNR, MSE, and FOM.

## 1. Introduction

Edge detection in grayscale images has been a well-established area of research for several decades, with algorithms that are now considered relatively mature. However, with the rapid advancements in technology and computer science, the use of color images has become more widespread, sparking increased interest in color image processing in recent years. Color space processing has found diverse applications, including color style transfer [1], color image segmentation [2], and color image encryption [3], to name just a few. In color images, edges—representing abrupt color transitions at the pixel level—are key low-level features. While traditional edge detection techniques, such as Roberts [4], Prewitt [5], and Sobel [6], work effectively for grayscale images, color images contain richer visual information that can significantly enhance edge detection. Existing methods for detecting edges in color images, when incorporating different color spaces and filters, can be generally classified into three categories: vector-based output synthesis, filter-driven approaches, and component decomposition and synthesis.

Edge detection in color images has garnered increasing attention due to its vital role in various computer vision tasks such as object recognition, image segmentation, and scene analysis. While traditional methods have been effective for grayscale images, the richer color information in color images presents additional challenges and opportunities for enhancing edge detection performance. These challenges have motivated the exploration of advanced techniques in the field, including deep learning approaches, which have set new benchmarks in recent years. However, deep learning methods often rely heavily on large labeled datasets and substantial computational resources, making them less accessible for resource-constrained applications. Consequently, there remains a need for robust edge detection methods that effectively leverage both traditional and advanced techniques, striking a balance between accuracy and computational efficiency.

The first class of edge detection methods involves vector output synthesis, where pixels are expressed as vector fields in various color space representations. These methods use one-dimensional or multi-dimensional responses and apply Euclidean or cosine distance metrics to measure pixel distances. Edge maps are then constructed by analyzing the edges detected with different operators. For example, Yugander et al. [7] introduced a combined index for color image edge detection that improves edge stability by incorporating angle and amplitude measurements. Huang et al. [8] fused mathematical morphology with vector methods, incorporating vector color differences in edge detection, although at the cost of excessive smoothing and loss of fine details. Similarly, Arbelaez et al. [9] utilized generalized features to capture both local and global contour information, enhancing edge contour detection but leading to some blurring and brightness loss.

The second approach is based on filter-based methods. Canny et al. [10] pioneered the concept of preprocessing images with filters before edge extraction, a technique that remains foundational in contemporary edge detection. Tian et al. [11] proposed a fusion edge detection method based on an enhanced Sobel operator, combining wavelet transform filtering with various operators to improve edge detection accuracy. However, this approach can sometimes result in the loss of fine edge details. Sun et al. [12] extended the Canny operator to color images by applying a Gaussian filter for smoothing, followed by first-order and higher-order weighted averaging for edge detection. While effective, this method is particularly sensitive to noise due to its reliance on single-scale operations. Kumar et al. [13] introduced the BM3D filter into color imaging to enhance noise robustness while preserving image details and polarization characteristics.

The third approach, component decomposition and synthesis, decomposes color images into individual channels, applies edge detection to each channel, and then synthesizes the results. Han et al. [14] proposed this method, where the image is first decomposed into three color channels, and edge detection is performed on each channel separately before the results are fused. Dollár et al. [15] incorporated random forests and decision tree theory with multiscale detection and edge sharpening to improve edge quality, though the method still faces challenges in capturing detailed edge information. Wang et al. [16] combined anisotropic properties with Gaussian directional derivatives to perform three-channel decomposition for edge detection, showing good robustness to noise [17,18,19,20].

In this paper, we propose a novel color image edge detection algorithm based on multiscale gradient fusion using color BM3D filters. The specific innovation points can be divided into the following parts:**●** The proposed algorithm combines multiscale gradient fusion with color BM3D filters for edge detection, harnessing the benefits of sparse coding and image denoising to significantly improve edge quality.**●** By integrating the XYZ color space with anisotropic directional derivatives at multiple scales, a novel edge strength map (ESM) is generated, enhancing both the accuracy and robustness of edge detection.**●** The ESM is optimized through pixel-average fusion and morphological filling techniques, effectively balancing large-scale edge elongation with small-scale resolution accuracy to achieve improved edge detection performance.**●** The proposed algorithm is evaluated using standard metrics (PR curve, AUC, PSNR, MSE, FOM) and compared with leading edge detection methods, including Color Sobel, Color Canny, SE, and Color AGDD, showcasing its superior performance on both edge and non-edge detection datasets.

The structure of this paper is as follows: Section 2 provides an introduction to the RGB to XYZ color space conversion and discusses the Colored Block-Matching and 3D (CBM3D) filter. Section 3 presents the multiscale gradient fusion edge detection algorithm, along with the construction of the fused ESM. Section 4 compares the proposed method to other existing techniques, conducts ablation experiments, and analyzes the results. Finally, Section 5 concludes the paper with a summary of findings and suggestions for future work.

## 2. Related Work

In this section, the CIE XYZ color space was introduced, and the RGB to XYZ space conversion was calculated. Then, the color block-matching and 3D (CBM3D) filter is presented.

### 2.1. CIE XYZ Color Space

The CIE XYZ color space, defined by the International Commission on Illumination (CIE), is an ideal color model based on the three primary colors—red, green, and blue [21,22,23]. It was the first color space to be mathematically formulated [24,25]. In the XYZ color space, the three components (X, Y, and Z) do not directly correspond to the physiological responses of human vision to short, medium, and long wavelengths, but rather represent a linear combination of the color-matching functions that roughly align with red, green, and blue [26,27,28]. Specifically, the Y component, derived from the green tristimulus value, represents brightness, while the X and Z components correspond to the red and blue stimuli, respectively. These three primary components can be used to represent all perceivable colors with positive values.

Unlike the RGB color space, which relies on physical devices for brightness information, the XYZ color space incorporates brightness, with the Y component directly encoding it. In contrast, the RGB color space requires specific instruments for this calculation, while brightness in the XYZ space can be derived mathematically.(1)$$\left[\begin{array}{c} X\\ Y\\ Z \end{array}\right]=\left[\begin{array}{ccc} 0.412453 & 0.357580 & 0.180423\\ 0.212671 & 0.715160 & 0.072169\\ 0.019334 & 0.119193 & 0.950227 \end{array}\right]\left[\begin{array}{c} R\\ G\\ B \end{array}\right]\;$$

As shown in Equation (Equation 1), Z=0.019334·R+0.119193·G+0.950227·B, and the arithmetic sum of the three coefficients equal 1.088754, which is close to the integer 1. Assuming the coefficients sum to 1, the values of X, Y, and Z can be constrained to the range [0, 255]. By manually adjusting the sum of the coefficients to 1, the mapping between the XYZ and RGB color spaces can be made equivalent, ensuring both spaces have the same value range. However, based on the tristimulus values in the XYZ color space, there is a forward transformation of the XYZ and L·a·b color spaces. The specific formula is shown in Formula (2).(2)$$\begin{array}{cc} L & =116f\left(\frac{Y}{Y_{n}}\right)-16\\ a & =500\left[f\left(\frac{X}{X_{n}}\right)-f\left(\frac{Y}{Y_{n}}\right)\right)\\ b & =200\left[f\left(\frac{Y}{Y_{n}}\right)-f\left(\frac{Z}{Z_{n}}\right)\right] \end{array}$$
where(3)$$f\left(t\right)=\left\{\begin{array}{cc} t^{1/3} & \mathrm{if}\;t>{\left(\frac{6}{29}\right)}^{3}\\ \frac{1}{3}{\left(\frac{29}{6}\right)}^{2}t+\frac{16}{116} & \mathrm{if}\;t\leq {\left(\frac{6}{29}\right)}^{3} \end{array}\right.\;$$

In Formula (3), the f(t) function has two domains to prevent infinite slope when *t* equals zero. At $t=t_{0}$, f(t) is a linear function, that is $t_{0}^{1/3}=at_{0}+b$, and its matching slope satisfies $a=\frac{1}{3t_{0}^{2/3}}$. Figure 1 shows the difference in edge detection between XYZ, RGB, L·a·b and HSV color spaces.

Among the different color spaces, the HSV and RGB color spaces are more vivid and bright, while the XYZ chromaticity space is more complete, and the L·a·b space appears relatively blurred. When expanding the components of the XYZ and RGB color spaces, X, Y, and Z tend to be brighter than R, G, and B, and the color differences are not significant. In edge detection results across various color spaces, the RGB color space shows a tendency to miss details and have incomplete edges. On the other hand, the L·a·b and HSV color spaces exhibit some jaggedness and slight edge stretching, which can be observed in the green oval box. After edge detection, RGB, L·a·b, and HSV color spaces suffer from varying degrees of edge loss, as indicated by the red circle in Figure 1. However, the XYZ color space performs better in this regard. Comparatively, L·a·b produces false edges, while HSV generates more false edges, whereas the other two color spaces perform well, as highlighted in the blue rectangle.

### 2.2. The Color Block-Matching and 3D Filter (CBM3D)

An image in the XYZ color space with noise variance, $\varepsilon_{\mathrm{dB}}^{2}$ can be modeled as $\phi_{\mathrm{XYZ}}=I_{\mathrm{XYZ}}+\mu_{\mathrm{XYZ}}$, where $I_{\mathrm{XYZ}}=\left[I_{X},I_{Y},I_{Z}\right]$ is actual image or input image and $\mu_{\mathrm{XYZ}}=\left[\mu_{X},\mu_{Y},\mu_{Z}\right]$ is represented as multiplicative or additive noise. Following the existing BM3D algorithm [29,30], the color block-matching and 3D (CBM3D) algorithm can be introduced. First, the XYZ color space image is transformed into a luminance–chrominance space, represented by the Y, U, and V channels. Among these, the U and V channels predominantly contain low-frequency data, while the Y channel, which has the highest signal-to-noise ratio (SNR), carries the most valuable information, including texture, edges, and lines. Next, the 2D images of these channels are divided into groups, and the resulting blocks are represented as a 3D array. After the transformation, a hard threshold is applied to attenuate noise and shrink the transform spectrum. Finally, an inverse 3D transform is used to reconstruct the grouped blocks. A weighted average of the underlying estimated block channels is computed and aggregated to produce block-by-block estimates for the corresponding Y, U, and V channels. These estimates are then converted back into the XYZ color space.

The algorithm for denoising using the CBM3D filter mainly has the following four steps.

(1) Channel conversion. $\phi_{\mathrm{XYZ}}$ is transformed into $\phi_{\mathrm{YUV}}$, and $\phi_{\mathrm{YUV}}=\left[\phi_{Y},\phi_{U},\phi_{V}\right]$ where the transformation matrix and opponent transformation satisfies. The specific formula is shown in Formula (4).(4)$$T=\left[\begin{array}{ccc} 0.3 & 0.59 & 0.11\\ -0.17 & -0.33 & 0.5\\ 0.5 & -0.42 & -0.08 \end{array}\right]\;T_{\mathrm{op}}=\left[\begin{array}{ccc} \frac{1}{3} & \frac{1}{3} & \frac{1}{3}\\ \frac{1}{\sqrt{6}} & 0 & -\frac{1}{\sqrt{6}}\\ \frac{1}{3\sqrt{2}} & -\frac{2}{3\sqrt{2}} & \frac{1}{3\sqrt{2}} \end{array}\right]$$

(2) Collaborative hard thresholding and chroma grouping constraints are used for basic estimation, denoted by $I_{\mathrm{YUV}}^{\mathrm{basic}}=\left[{\dot{I}}_{Y}^{\mathrm{basic}},{\dot{I}}_{U}^{\mathrm{basic}},{\dot{I}}_{V}^{\mathrm{basic}}\right]$. For each block, a 2D Kaiser window is used for $\phi_{Y}$ and block estimation aggregation, and a 3D numerical permutation matrix is obtained from the stacked blocks located at the $\phi_{Y}$, $\phi_{U}$, and $\phi_{V}$ positions.

(3) Based on $I_{\mathrm{YUV}}^{\mathrm{basic}}$. The collaborative Wiener filtering and chroma grouping constraints are used to obtain final estimates. Block matching is used for finding the locations of block in ${\dot{I}}_{Y}^{\mathrm{basic}}$, and then a set of 3D arrays are constructed by stacking blocks in ${\dot{I}}_{Y}^{\mathrm{basic}}$ and $\phi_{Y}$. Consequently, a 3D transformation is applied to these arrays, followed by 3D Wiener filtering on the noisy arrays using the energy spectrum derived from the base estimate.

(4) All grouped block estimates are calculated by inverse 3D transform. Additionally, the block estimates of the corresponding Y, U, and V channels are calculated by average weighting, denoted ${\dot{I}}_{Y}^{\mathrm{final}},{\dot{I}}_{U}^{\mathrm{final}},{\dot{I}}_{V}^{\mathrm{final}}$.

## 3. A New Color Image Edge Detection Method

After denoising with the CBM3D filter, the XYZ color space images are used to compute their color vector gradients and anisotropic Gaussian directional derivative gradients. Next, a pixel-by-pixel average fusion of multiple gradients is performed to generate a new edge strength map. Simultaneously, the edge map is normalized, and non-maximum suppression is applied. Double-threshold selection and morphological fill refinement are then utilized to extract the edge contours. Finally, a novel color image edge detection method is proposed.

### 3.1. Multiscale Gradient Fusion Approach

Assume that the input image in the XYZ color space is $I_{\mathrm{xyz}}\left(x\right)$, where $x={\left[\begin{array}{c} u,vs. \end{array}\right]}^{T}$, and the scale is σ. The expression of the two-dimensional Gaussian kernel function can be defined as Formula (5).(5)$$\begin{array}{cc} G_{\sigma }\left(x\right) & =\frac{1}{2\pi \sigma^{2}}\exp\left(\frac{-x^{T}x}{2\sigma^{2}}\right)\\ \; & =\frac{1}{2\pi \sigma^{2}}\exp\left(-\frac{u^{2}+v^{2}}{2\sigma^{2}}\right)\\ \; & =\frac{1}{\sqrt{2\pi }\sigma }\exp\left(-\frac{u^{2}}{2\sigma^{2}}\right)\times \frac{1}{\sqrt{2\pi }\sigma }\exp\left(-\frac{v^{2}}{2\sigma^{2}}\right)\\ \; & =G_{\sigma }\left(u\right)\times G_{\sigma }\left(v\right) \end{array}$$
where *U* and *V* are the coordinates of the pixel point, and *T* represents the transpose of the matrix. For XYZ color images, convolve with a Gaussian function and calculate the gradient magnitude. The concrete process is like the Formulas (6)–(8).(6)$$I_{\mathrm{xyz},\sigma }\left(x\right)=I_{\mathrm{xyz}}\left(x\right)\cdot G_{\sigma }\left(x\right)=\left[\begin{array}{c} \;\int \int \;I_{x}\left(x-\delta \right)G_{\sigma }\left(\delta \right)\;d\delta \\ \;\int \int \;I_{y}\left(x-\delta \right)G_{\sigma }\left(\delta \right)\;d\delta \\ \;\int \int \;I_{z}\left(x-\delta \right)G_{\sigma }\left(\delta \right)\;d\delta \end{array}\right]$$(7)$$\nabla I_{\mathrm{xyz},\sigma }\left(x\right)=\left[\begin{array}{c} \frac{\partial }{\partial u}I_{\mathrm{xyz},\sigma }\left(x\right)\\ \frac{\partial }{\partial v}I_{\mathrm{xyz},\sigma }\left(x\right) \end{array}\right]=\left[\begin{array}{c} I_{x}\cdot \nabla G_{\sigma }\left(x\right)\\ I_{y}\cdot \nabla G_{\sigma }\left(x\right)\\ I_{z}\cdot \nabla G_{\sigma }\left(x\right) \end{array}\right]$$
where(8)$$\nabla G_{\sigma }\left(x\right)=\frac{-x}{\sigma^{2}}\cdot G_{\sigma }\left(x\right)=\frac{-x}{2\pi \sigma^{4}}\exp\left(\frac{-x^{T}x}{2\sigma^{2}}\right)$$

After computing the gradient magnitude for each color space, the composite form is obtained by combining the results from each plane and their individual contributions. The concrete process is like the Formulas (9) and (10).(9)$$\nabla P\left(x\right)=\nabla P_{XY}\left(x\right)+\nabla P_{YY}\left(x\right)+\nabla P_{ZY}\left(x\right)$$(10)$$\begin{array}{cc} \nabla P_{XY}\left(x\right) & =\nabla P_{XY}\left(u,v\right)=\sqrt{{\left[I_{x}\cdot \nabla G_{\sigma }\left(u\right)\right]}^{2}+{\left[I_{x}\cdot \nabla G_{\sigma }\left(v\right)\right]}^{2}}\\ \nabla P_{YY}\left(x\right) & =\nabla P_{YY}\left(u,v\right)=\sqrt{{\left[I_{y}\cdot \nabla G_{\sigma }\left(u\right)\right]}^{2}+{\left[I_{y}\cdot \nabla G_{\sigma }\left(v\right)\right]}^{2}}\\ \nabla P_{ZY}\left(x\right) & =\nabla P_{ZY}\left(u,v\right)=\sqrt{{\left[I_{z}\cdot \nabla G_{\sigma }\left(u\right)\right]}^{2}+{\left[I_{z}\cdot \nabla G_{\sigma }\left(v\right)\right]}^{2}} \end{array}$$

The result in the composite form is compared with a threshold to obtain the final color gradient magnitude. For Formula (5), referring to the anisotropic factor ρ, the Gaussian function can be deformed to obtain the anisotropic Gaussian kernel function. The concrete process is like the Formulas (11) and (12).(11)$$G_{\sigma ,\rho ,\theta }\left(x\right)=\frac{1}{2\pi \sigma^{2}}\exp\left[\frac{-1}{2\sigma^{2}}\left(x^{T}R_{-\theta }\left[\begin{array}{cc} \rho^{2} & 0\\ 0 & \rho^{-2} \end{array}\right]\left(R_{\theta }x\right)\right)\right]$$(12)$$R_{\theta }=\left[\begin{array}{cc} \cos\theta & \sin\theta \\ -\sin\theta & \cos\theta \end{array}\right]$$
where $R_{\theta }$, a symbol representing the direction of rotation without affecting the result, denotes the rotation matrix, and θ is the rotation angle. To obtain the anisotropic Gaussian directional derivative (AGDD), we differentiate the anisotropic Gaussian kernel function. The details are shown in Formula (13) below.(13)$$G_{\sigma ,\rho ,\theta }^{\prime }\left(x\right)=\frac{(\cos\theta ,\sin\theta )x}{\sigma^{2}\rho^{-2}}G_{\sigma ,\rho ,\theta }\left(x\right),\;\theta =\frac{(N-1)\pi }{n},\;n=1,2,3,\ldots ,N$$

*N* represents the number of directions for the anisotropic Gaussian directional derivative (AGDD). Based on Equations (5) and (13), the color gradient and the anisotropic Gaussian directional derivative gradient for the multiscale parameter are fused. Their average gradient value is computed to generate a new edge strength map (ESM). The concrete process is like the Formula (14).(14)$$\mathrm{ESM}=\max_{n=1,2,3,\ldots ,N}\left\{\left|\frac{\left|\nabla P(x)\right|+\left|I_{\mathrm{xyz}}\cdot G_{\sigma_{1},\rho ,\theta }^{\prime }\left(x\right)\right|+\left|I_{\mathrm{xyz}}\cdot G_{\sigma_{2},\rho ,\theta }^{\prime }\left(x\right)\right|}{3}\right|\right\}$$

The image in the XYZ color space is continuously represented, enabling a more precise analysis of its color components. After computing gradients at multiple scales, the gradient maps are fused by averaging pixel-by-pixel, effectively combining complementary scale information while reducing noise. The maximum gradient value at each pixel is then selected to create the Edge Strength Map (ESM), emphasizing the most significant edges. To ensure consistency across images, the ESM is normalized by dividing each pixel by the map’s largest value, standardizing the range and improving the accuracy and stability of subsequent edge detection steps.

The proposed method employs a multiscale framework for extracting edge strength maps (ESMs), where each scale offers complementary advantages: the small-scale ESM provides high edge resolution and accuracy but is prone to noise sensitivity; the large-scale ESM effectively suppresses noise, though it may cause excessive edge elongation; and the intermediate scale strikes a balance between resolution, accuracy, and noise robustness. By leveraging scale fitting techniques to combine these scales, the algorithm synthesizes a unified edge map (EM) that integrates the strengths of each scale while mitigating their respective limitations. This approach ensures enhanced adaptability to varying noise levels and image characteristics.

### 3.2. Color Image Edge Detection Process

Given a color image $I_{\mathrm{RGB}}\left(x\right)$, the XYZ space image is derived using Formula (1), along with the specified parameters σ, ρ and *N*. The color space gradient and the AGDD gradient are then computed using the Formulas (9)–(11). The flowchart for the multiscale gradient fusion edge detection algorithm, based on the CBM3D filter, is shown in Figure 2.

To achieve the effective denoising of color images, the CBM3D filter is applied for filtering. A new edge strength map (ESM) is then constructed using a multiscale gradient approach and gradient vector fusion. Finally, an optimized edge map is obtained. The specific edge detection process is outlined as follows, as illustrated in the figure above.

(1) XYZ Color Space Conversion and Filtering: Initially, the input RGB image is converted to the XYZ color space using Formula (1). Subsequently, channel conversion is performed as described in Formula (4), and the grouped block estimates are constructed. After applying 3D Wiener filtering, the estimated values for each block in the channel are obtained through a 3D inverse transformation followed by weighted averaging. Finally, the image is converted back to the XYZ color space.

(2) Average fusion for the New ESM: After applying CBM3D filtering, the vector gradient of the color image is computed using Formulas (9) and (10). Next, the anisotropic Gaussian directional derivative (AGDD) is calculated as per Formula (13), and the gradients for two scales are obtained by convolving the image with the corresponding filters. Finally, the three gradients are averaged to generate the new edge strength map (ESM).

(3) Normalization and non-maximum suppression. Based on the ESM, the largest pixel value is identified and used to normalize the image by dividing all pixel values by this maximum. Then, using the modulus value of the ESM and its gradient direction, pixels are retained if the gradient magnitude along any side of the pixel, in the direction of the gradient, exceeds the pixel’s magnitude; otherwise, the pixel is set to zero. This process results in a set of local maxima, which is commonly referred to as the candidate edge pixel set, denoted by $\Lambda_{\max}$.

(4) Double threshold selection. This step is essential for effective edge detection. Without it, the detected edges would only form feature maps, rather than distinct contour lines. Double thresholds named high threshold $H_{\mathrm{th}}$ and low threshold $L_{\mathrm{th}}$ can be calculated as (15) and (16).(15)$$\phi \left\{x:\Lambda_{max}\left\{x\right\}<H_{\mathrm{th}}\right\}=\lambda M$$(16)$$\phi \left\{x:\Lambda_{max}\left\{x\right\}<L_{\mathrm{th}}\right\}=\mu \lambda M$$
where φ represents the set of finite candidate sets, *M* represents the image resolution size, λ is the parameter for calculating the high threshold, which ranges within the interval [0.6,0.95], and 0.4≤μ≤0.6. After non-maximum suppression, for the set $\Lambda_{\max}$, edge pixels with values greater than the threshold λ are classified as strong edge pixels, denoted by $S_{\mathrm{edge}}$. In the eight-neighborhood range of $W_{\mathrm{edge}}$, if a path is connected to the strong edge pixel $S_{\mathrm{edge}}$, these pixels are regarded as edge pixels, denoted as $WS_{\mathrm{edge}}$. The final edge contour map is then composed of the combined sets $W_{\mathrm{edge}}$, $W_{\mathrm{edge}}$, and $WS_{\mathrm{edge}}$.

(5) Morphological refinement and filling. To obtain finer edges that are closer to the ground truth (GT) image, a morphological optimization operation is applied. Unlike traditional methods that focus on edge thinning, the approach used here involves padding isolated inner voxels in the Edge Strength Map (ESM), where pixels with a value of 0 are filled with 1. This helps to enhance the continuity of edges. Finally, the refined edge map is produced as the output.

## 4. Experimental Results and Performance Analysis

In this section, the key findings of this study are presented. For simplicity, this section is divided into three. This section includes the experimental environment, the measurement indicator, and the analysis of results.

### 4.1. Experimental Results

To assess the performance and benefits of the color image edge detection algorithm using the CBM3D filter for multiscale gradient fusion, we compare the proposed method with several existing algorithms, including Color Sobel, Color Canny, SE, and Color AGDD. The comparison is performed on both edge and non-edge image sets. The edge detection datasets, such as BIPED and MDBD, include original RGB images as well as images modified through enhancement techniques like inversion translation. The non-edge datasets are sourced from the well-known BSDS500 dataset, the PASCAL 2007 challenge dataset, and natural scene images. To ensure a fair evaluation, the parameters for each algorithm are configured as follows.

(1) The color Sobel algorithm is manipulated in the RGB color space. The templates used 3 × 3 and 5 × 5. The 5 × 5 templates are given as follows:$$\left[\begin{array}{cccc} 0 & 0 & 0 & 1\\ 0 & 1 & 1 & 0\\ 1 & 1 & 1 & 1\\ 0 & 1 & 1 & 0\\ 0 & 0 & 0 & 1 \end{array}\right]$$

(2) The color Canny algorithm. It uses the RGB color space, and the scale parameter is selected as $\sqrt{2}$.

(3) The SE algorithm adopts the RGB color space, and adds edge sharpening and thinning operations to optimize the edge effect.

(4) The color AGDD algorithm is completed at the RGB color space, and the scale and anisotropic factor are both 7, and the direction number is 16.

(5) The two scale parameters of the proposed algorithm are selected as $\sqrt{3}$ and $\sqrt{7}$.

The XYZ color space is used for images without noise, and the RGB color space is used for noise images. Figure 3 presents five representative images selected from four datasets and natural scene images, along with corresponding zero-mean Gaussian white noise images with a noise density of 0.01, and their ground truth (GT) maps. For images without GT maps, results are obtained through the use of five different algorithms. (Note that a noise variance of 25.5 is numerically equivalent to a noise density of 0.01).

The images in the five datasets feature rich colors that broadly represent the diversity of available images. Additionally, they contain high-level semantic information, such as facial details, low-frequency sky information, and high-frequency object features. Figure 4 displays the detection results of the five algorithms on these images without noise.

As shown in Figure 4, the Color Sobel algorithm detects the general outlines of the images, but the results are cluttered due to the lack of smoothing filters. The Color Canny algorithm performs well in edge detection, aided by its Gaussian filter, particularly in low-frequency spatial images. However, using an isotropic filter causes some loss of detail, as it sacrifices image resolution or noise robustness to preserve edge information. The SE algorithm, though fast in detection, is significantly disturbed by noise, even with its multiscale factors and edge-sharpening operations. The Color AGDD algorithm performs better on face and object feature images but introduces false edges in low-frequency spatial images. In contrast, the proposed algorithm successfully detects edges in the noise-free images, showing superior performance in handling both low-frequency and high-level information. To evaluate the algorithm’s robustness to noise, a zero-mean Gaussian white noise with a density of 0.01 was added in subsequent experiments. The edge detection results of various algorithms under noisy conditions are shown in Figure 5.

For images with Gaussian white noise, the Color Sobel and SE algorithms are heavily affected by noise, resulting in a large number of stains. However, they have the fastest detection speeds. The Color Canny algorithm performs slightly better but remains less robust to high-density noise. The Color AGDD algorithm shows some noise robustness but loses edge details due to its strong smoothing filter. This is particularly evident in the “airplane” image, which contains low-frequency information, where false edges caused by noise appear. In contrast, the proposed algorithm performs well on both low-frequency and high-frequency images, such as face images, with accurate and detailed edge detection. Its robustness to noise can be attributed to the effective filter performance and the use of multiscale gradients.

For images with Gaussian white noise, both the Color Sobel and SE algorithms exhibit a large number of false edges, heavily influenced by noise. However, they offer the fastest detection speeds. The Color Canny algorithm performs slightly better, with fewer false edges, though it is less robust to high-density noise. The Color AGDD algorithm is more robust to noise but tends to lose edge details due to its strong smoothing filter, especially in aircraft images with low-frequency information, where false edges caused by noise are more prominent. In contrast, the proposed algorithm demonstrates excellent performance in both low-frequency and high-frequency face images, producing accurate and detailed edge detections. Its strong noise robustness is attributed to the effective filter design and the use of multiscale gradients.

### 4.2. Performance Analysis

To quantitatively evaluate five edge detection algorithms, the PR curve, AUC, PSNR, MSE, and FOM index are employed as performance metrics. Initially, the PR curve and AUC are used to assess the algorithm’s accuracy. In the PR curve, the x axis represents precision, while the y axis represents recall. Precision is defined as the proportion of correctly predicted positive edges among all predicted positive edges. Recall, on the other hand, reflects the proportion of true positive edges that are correctly identified. The precision and recall are calculated as follows. The details are shown in Formulas (17) and (18).(17)$$L_{precision}=\frac{\eta_{\mathrm{TP}}}{\eta_{\mathrm{TP}}+\eta_{\mathrm{FP}}}$$(18)$$L_{recall}=\frac{\eta_{\mathrm{TP}}}{\eta_{\mathrm{TP}}+\eta_{\mathrm{FN}}}$$
where, $\eta_{\mathrm{TP}}$ is the edge pixel of the true positive, $\eta_{\mathrm{FN}}$ is the edge pixel of the false negative, and $\eta_{\mathrm{FP}}$ indicates that it is not an edge but it is detected. In addition, AUC is defined as the area under PR curve, which can quantitatively reflect the algorithm performance measured based on the PR curve. The value of AUC can be integrated along the horizontal axis by the PR curve. The details are shown in Formula (19).(19)$$\mathrm{AUC}=\sum_{i=1}^{N_{\mathrm{recall}}}\frac{\left(L_{\mathrm{precision}}\left(i\right)+L_{\mathrm{precision}}\left(i+1\right)\right)\left(L_{\mathrm{recall}}\left(i+1\right)-L_{\mathrm{recall}}\left(i\right)\right)}{2}$$

The experiments were conducted on several benchmark datasets, employing a uniform threshold of 0.001 within the range [0, 1] for the positive sequence values. Precision–recall (PR) curves were generated for five different edge detection algorithms, and the results are illustrated in Figure 6 and Figure 7. To further quantify the performance of each algorithm, the area under the curve (AUC) values were computed using Formula (19) and are clearly indicated in the figures. These AUC values provide a comprehensive measure of the algorithms’ ability to distinguish between positive and negative sequences across various thresholds, offering a detailed comparison of their overall effectiveness in edge detection tasks.

The straight line y = x in the figure serves as a reference for performance comparison. In the absence of noise, the five algorithms show minimal differences in their results, with the proposed method achieving the best performance and highest accuracy. Specifically, the AUC value for the proposed algorithm reaches 0.98622, which is 1.833% higher than that of the SE algorithm. However, when noise is introduced, a significant performance gap emerges among the algorithms. The Color Sobel algorithm demonstrates the poorest performance, as it is highly sensitive to noise, yielding an AUC value of only 0.91942. In contrast, the proposed algorithm maintains strong performance under noisy conditions, with an AUC value of 0.98367, which is 6.425% higher than that of the Color Sobel method. To further assess and compare the performance of the algorithms, additional evaluation metrics, such as peak signal-to-noise ratio (PSNR) and mean squared error (MSE), are incorporated. These metrics provide a more comprehensive understanding of the accuracy and robustness of the algorithms in different scenarios. The PSNR and MSE values are calculated using Formulas (20) and (21), respectively, and offer valuable insights into the algorithms’ ability to preserve image quality and minimize error during edge detection.(20)$$\mathrm{PSNR}=10\times \log_{10}\left(\frac{{\left(2^{\tau }-1\right)}^{2}}{\mathrm{MSE}}\right)$$(21)$$\mathrm{MSE}=\frac{1}{H\times W}\sum_{i=1}^{H}\sum_{j=1}^{W}\left(I_{\mathrm{detected}}\left(i,j\right)-I_{\mathrm{GT}}\left(i,j\right)\right)$$

In Formula (20), τ represents the bit value of each pixel, which is generally 8, so that the gray level of the pixel is 256. For Formula (21), *H* and *W* represent the height and width of the image, respectively. $I_{\mathrm{detected}}$ represents the output image of the algorithm to detect edges and $I_{\mathrm{GT}}$, the ground-truth image. According to the formula, the values of the five algorithms in the case of PSNR and MSE are obtained and filled in Table 1 and Table 2.

As shown in the data in Table 1, the SE algorithm exhibits the worst performance, with a PSNR of only 55.4221 and an MSE as high as 0.1880 on the Lena image. The PSNR and MSE values of the Color Sobel and Color Canny algorithms are similar, indicating comparable edge detection performance. The edges detected by the Color AGDD algorithm are of high quality, closely matching the ground-truth (GT) image and true edge levels. However, the proposed algorithm demonstrates superior performance on the PASCAL VOC 2007 dataset, achieving the highest PSNR of 60.5842 and the lowest MSE of 0.0573. This represents a PSNR improvement of 2.1935 over the SE algorithm and a 7.45% reduction in MSE on the Lena image.

Similarly, the Color Sobel and Color Canny algorithms perform poorly in the presence of noise, indicating that both are not robust to high levels of noise. Under the influence of intense noise, the average PSNR of the Color Sobel algorithm drops to 54.2675, showing a significant deviation from the GT image. The SE and Color AGDD algorithms exhibit minimal performance degradation under noisy conditions, maintaining competitive performance. However, the proposed algorithm stands out with an average PSNR of 58.8033, which is 4.5358 higher than that of the Color Sobel algorithm. In addition, the average MSE of the Color Sobel algorithm is 0.2463, which is 15.79% higher than the proposed algorithm.

Finally, the FOM (figure of merit) is used to evaluate the performance of the five algorithms. The FOM takes into account three key factors: false edge generation, inaccurate edge positioning, and edge loss, as well as noise robustness. Based on these criteria, the definition of FOM is given as follows as (22).(22)$$\mathrm{FOM}=\frac{\sum_{i=1}^{E_{\mathrm{detected}}}\left\{1/[1+\nu d^{2}\left(i\right)]\right\}}{\mathrm{Max}\left(E_{\mathrm{ideal}},E_{\mathrm{detected}}\right)}\;$$
where, ϑ represents the loss factor on the edge pixels, and the general constant is 0.25. Additionally, d(i) represents the Euclidean distance between the *i*-th detected edge pixel and the true edge pixel at the corresponding position. Moreover, $E_{\mathrm{ideal}}$ and $E_{\mathrm{detected}}$ represent, respectively, the number of pixels in the ideal edge map, and the number of 0 to 1, and the best edge map gets FOM = 1. Furthermore, the FOM values of the five algorithms in the two cases are shown in Table 3 and Table 4.

For several key reasons, we chose the precision–recall (PR) curve and the Figure of Merit (FOM) as the primary evaluation metrics, rather than SSIM and RMSE. Firstly, the PR curve and FOM are widely acknowledged in edge detection research as they effectively capture the essential performance aspects of edge detection. The PR curve evaluates the trade-off between precision and recall, providing insights into how the algorithm balances false positives and false negatives. In contrast, the FOM metric is specifically designed for edge detection, assessing edge localization accuracy and noise robustness. It quantifies the alignment between detected and true edges, making it particularly suitable for comparing performance under varying noise conditions.

Although SSIM and RMSE are valuable for evaluating image quality and pixel similarity, they are less suitable for edge detection tasks. SSIM measures the structural similarity of the entire image, which, while useful for overall image quality assessment, does not provide specific insights into the accuracy or localization of the detected edges. RMSE, which evaluates pixel differences, may include irrelevant non-edge regions, thus reducing its relevance to edge detection tasks. Therefore, we selected the PR curve and FOM as the most intuitive and relevant metrics to effectively highlight the performance of the proposed method, demonstrating its superior edge detection capability.

According to the data in Table 3, the edge maps detected by the SE algorithm show poor performance, with the lowest FOM value at 0.9027 and only 796,326 edge pixels detected, resulting in significant edge detail loss. The FOM values for the Color Sobel and Color Canny algorithms are roughly similar, and their detected edge quality is also comparable. The Color AGDD algorithm performs best on the low-frequency “Airplane” image, achieving an FOM value of 0.9867 and detecting 159,350 edge pixels. However, many of these detected edges are false positives. In contrast, the proposed algorithm achieves the highest FOM value of 0.9873, which is very close to the ground truth image. When compared to the SE algorithm, the proposed algorithm’s average FOM is 3.698% higher.

### 4.3. Analysis of the Experimental Results

When noise is introduced, the performance of all algorithms decreases to varying extents, with corresponding fluctuations in the Figure of Merit (FOM) values. Among the algorithms tested, the Color Sobel method is the least robust under high-intensity noise, yielding an average FOM value of 0.82676 and detecting 351,487 edge pixels. Other algorithms show some advantages depending on the specific dataset, but their FOM values remain relatively similar.

In contrast, the proposed method demonstrates a superior performance, particularly in the low-frequency space, where it achieves a remarkable FOM value of 0.9871 and detects 159,993 edge pixels. The average FOM value of 0.973 is 14.624% higher than that of the Color Sobel algorithm and 1.808% higher than the next-best performing method, Color AGDD. Further experimental results underscore the effectiveness of the proposed algorithm, with outstanding AUC values of 0.98622 and 0.98367, outperforming existing methods in two distinct test scenarios. The algorithm also achieves the highest peak signal-to-noise ratio (PSNR) of 60.5842 and the lowest mean squared error (MSE) of 0.0573, further validating its robustness. Finally, the FOM evaluation solidifies the proposed method’s superiority in terms of edge detection accuracy when compared to state-of-the-art techniques.

## 5. Conclusions

This paper introduces a novel color edge detection method that combines the CBM3D filter with multiscale gradient fusion. The method begins with image segmentation to create a sparse matrix, followed by high-frequency noise removal using 3D transformations and collaborative filtering. Edge maps are generated through anisotropic directional derivatives and multiscale gradient vectors in the XYZ color space, and averaged to produce an Edge Saliency Map. The final edge map is refined through normalization, non-maximum suppression, double thresholding, and morphological processing. Experimental results show that the proposed method outperforms existing algorithms, demonstrating its robustness and effectiveness. Future work will explore hybrid approaches combining deep learning and traditional techniques to improve accuracy, robustness, and efficiency, with applications in dynamic video edge detection and real-time processing.

## Figures and Tables

**Figure 1 sensors-25-02031-f001:**
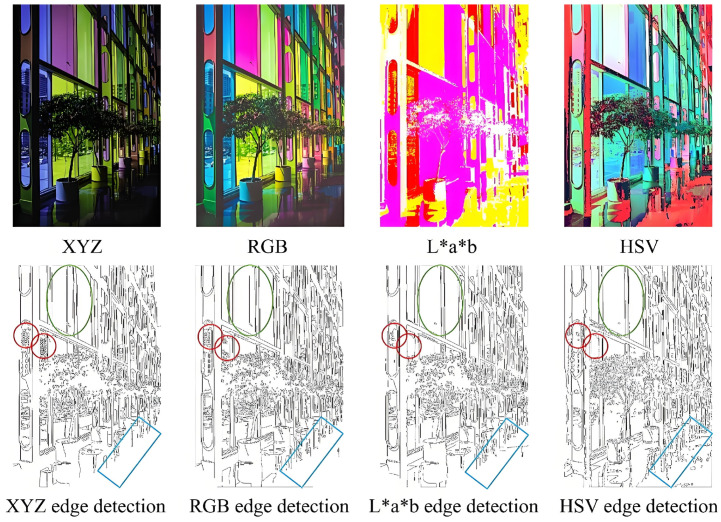
Four different color spaces and their corresponding edge detection.

**Figure 2 sensors-25-02031-f002:**
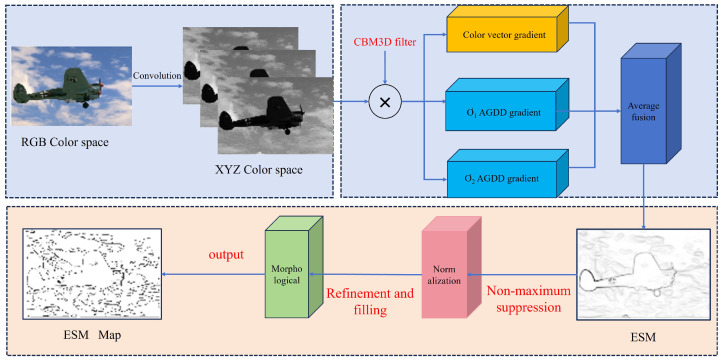
Flowchart of the color image edge detection via multiscale gradient fusion.

**Figure 3 sensors-25-02031-f003:**
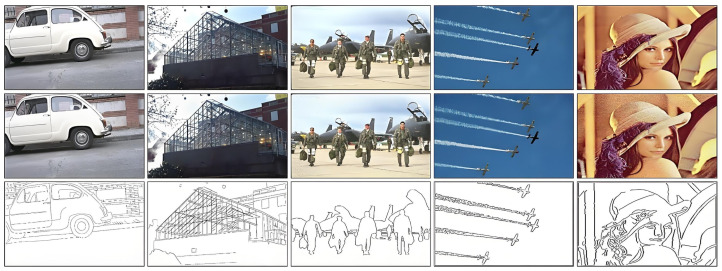
Different dataset images and their corresponding GT images.

**Figure 4 sensors-25-02031-f004:**
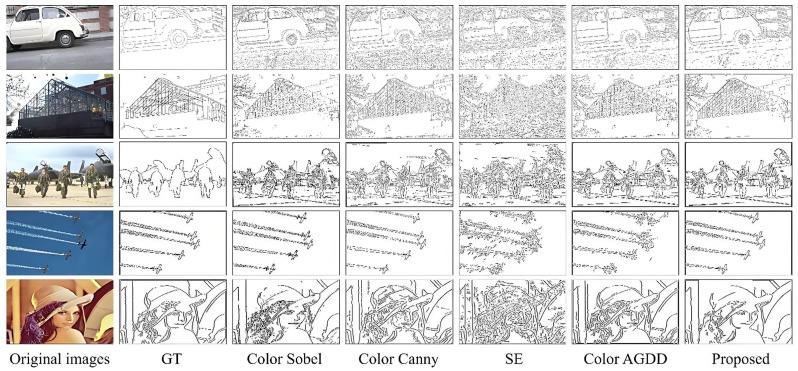
Results of different algorithms without noise.

**Figure 5 sensors-25-02031-f005:**
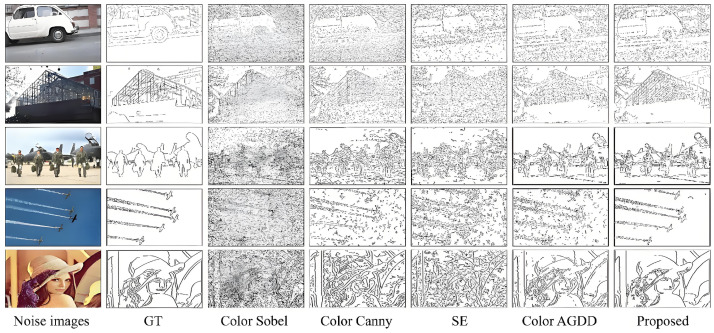
Results of different algorithms with a noise density of 0.01.

**Figure 6 sensors-25-02031-f006:**
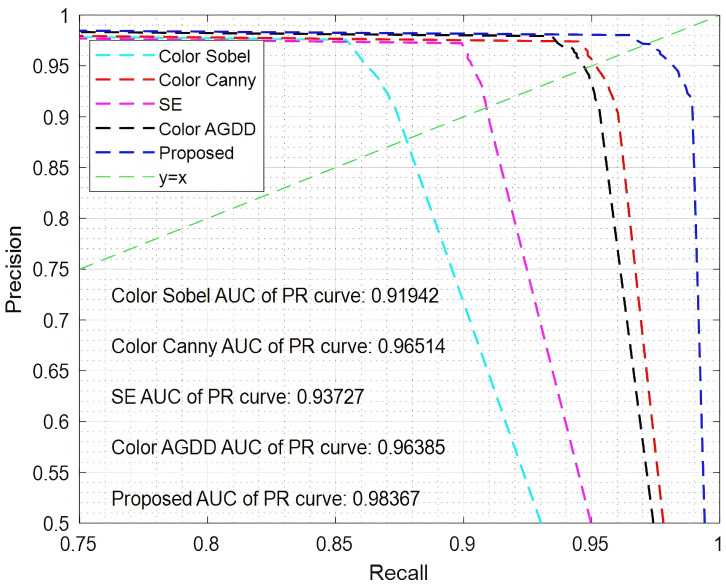
PR curve without noise.

**Figure 7 sensors-25-02031-f007:**
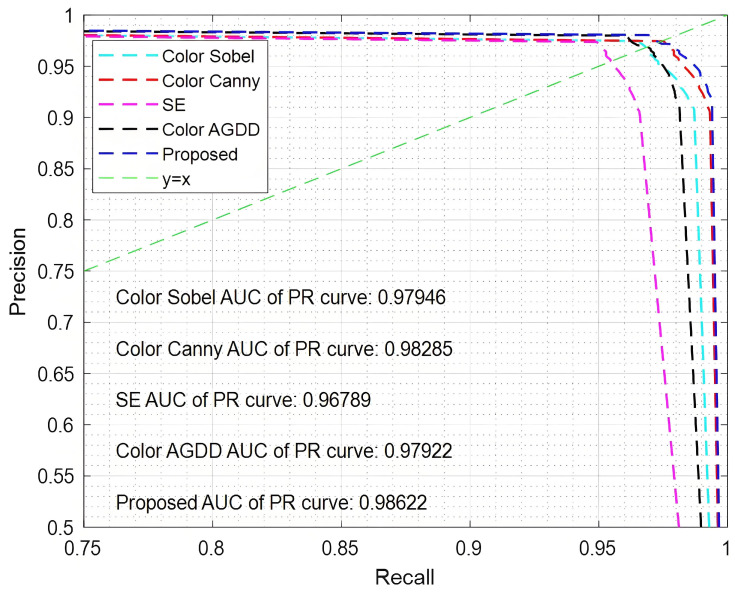
PR curve with a noise density of 0.01.

**Table 1 sensors-25-02031-t001:** PSNR and MSE values of the algorithms without noise.

PSNR/MSE	Images				
	Car	House	Airman	Aircraft	Lena
Color Sobel	58.3943/0.0949	57.9004/0.1063	57.8936/0.1064	59.4183/0.0749	56.6504/0.1417
Color Canny	57.7694/0.1095	57.3236/0.1214	58.4272/0.0941	59.7929/0.0687	56.5378/0.1454
SE	58.0964/0.1016	55.8104/0.1720	57.7789/0.1093	58.4296/0.0941	55.4221/0.1880
Color AGDD	59.3272/0.0765	57.6517/0.1125	58.5668/0.0912	59.7459/0.0696	57.5576/0.1150
**Proposed**	**59.7834/0.0689**	**58.0976/0.1016**	**59.0649/0.0813**	**60.5842/0.0573**	**57.6156/0.1135**

**Table 2 sensors-25-02031-t002:** PSNR and MSE values of the algorithms with noise density of 0.01.

PSNR/MSE	Images				
	Car	House	Airman	Aircraft	Lena
Color Sobel	54.4199/0.2369	54.5217/0.2314	54.7027/0.2219	54.0975/0.2551	53.5959/0.2470
Color Canny	56.4167/0.1496	56.6361/0.1422	57.5770/0.1145	58.2821/0.0973	55.7158/0.1445
SE	58.7171/0.1584	56.1659/0.1584	57.8254/0.1081	56.5962/0.1435	55.0543/0.1443
Color AGDD	57.8259/0.1081	57.6514/**0.1014**	58.9112/0.0842	58.3481/0.0959	57.2148/0.1435
**Proposed**	**59.1469/0.0798**	**58.0410**/0.1029	**59.1629/0.0795**	**60.2004/0.0626**	**57.4664/0.1174**

**Table 3 sensors-25-02031-t003:** FOM values of the algorithms without noise.

Images	Car	House	Airman	Aircraft	Lena
Color Sobel	0.9369/845722	0.9774/862113	0.9259/139695	0.9835/106130	0.9803/509976
Color Canny	0.9264/836760	**0.9927**/845996	0.9407/141995	0.9851/127551	0.9805/588674
SE	0.9321/841622	0.9027/796362	0.9324/139614	0.9847/158406	0.9447/566613
Color AGDD	0.9600/866915	0.9703/855833	0.9424/142501	0.9867/159053	0.9803/593848
**Proposed**	**0.9660/872163**	0.9832/**867293**	**0.9541/143970**	**0.9873/160691**	**0.9827/607784**

**Table 4 sensors-25-02031-t004:** FOM values of the algorithms with noise density of 0.01.

Images	Car	House	Airman	Aircraft	Lena
Color Sobel	0.7885/712008	0.8314/733244	0.8048/124160	0.9002/142312	0.8089/457610
Color Canny	0.8851/799586	0.9353/824910	0.9193/138744	0.9487/175579	0.9479/567214
SE	0.9472/855264	0.9137/809600	0.9259/139772	0.9484/150229	0.9524/522938
Color AGDD	0.9343/843733	0.9254/816180	0.9519/143666	0.9509/153138	0.9185/592720
**Proposed**	**0.9546/861797**	**0.9814/865615**	**0.9558/144224**	**0.9871/179133**	**0.9861/599494**

## Data Availability

Data will be made available upon request.

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
