# Peer review of "A Multiscale Gradient Fusion Method for Color Image Edge Detection Using CBM3D Filtering"

_sensors, 2025, doi:10.3390/s25072031_

Round 1

Reviewer 1 Report

Comments and Suggestions for Authors
  1. The paper does not compare the proposed method with any deep learning-based edge detection techniques. This omission makes it difficult to assess how the proposed method stacks up against the current state-of-the-art. Without such a comparison, the novelty and relevance of the proposed method in the context of modern edge detection research are unclear.
  2. While the proposed method introduces some novel elements (e.g., CBM3D filtering, multiscale gradient fusion, and morphological refinement) within the realm of traditional edge detection, these innovations are incremental rather than revolutionary. The core ideas (e.g., gradient-based edge detection, noise reduction) have been explored extensively in the past, and the proposed method primarily refines these ideas rather than introducing a fundamentally new paradigm.
  3.  Deep learning-based edge detection methods, such as HED (Holistically-Nested Edge Detection), RCF (Rich Feature Hierarchies for Edge Detection), and more recent approaches like PiDiNet and DexiNed, have set new benchmarks in edge detection. These methods leverage large datasets and powerful neural network architectures to learn complex edge features directly from data, often outperforming traditional methods in terms of precision and recall.
Comments on the Quality of English Language

1. Some sentences are overly long and complex, making them difficult to follow. For example:

"The Colored Block-Matching and 3D (CBM3D) filter is then applied to the sparse images to reduce noise interference, followed by the computation of vector gradients of the color image and the anisotropic Gaussian directional derivatives for two scale parameters, which are averaged pixel-by-pixel to generate a refined edge strength map."

2. Some phrases are ambiguous or could be rephrased for clarity. For example: "Edge features are further enhanced using image normalization and non-maximum suppression, followed by edge contour extraction through double-thresholding and a novel morphological refinement approach."

Reviewer 2 Report

Comments and Suggestions for Authors

This paper proposes a color image edge detection method based on CBM3D filtering and multi-scale gradient fusion, aiming to address the poor performance of traditional edge detection methods in noisy environments. The method adopted in this paper is reasonable, and its effectiveness has been demonstrated through quantitative experiments; the structure of the paper is also well-organized. The following are suggestions for improvement:

1.The introduction could include a description of the importance and necessity of this research.
2.How does the proposed multi-scale gradient fusion method maintain consistent performance across datasets with different noise characteristics? Are there specific edge cases in which this algorithm performs poorly compared to existing detectors?
3.Although the PR curve is used as an evaluation metric, should other metrics (such as Structural Similarity Index Measure (SSIM) or Root Mean Square Error (RMSE)) be considered to further understand the edge detection performance, especially under noisy conditions?
4.Some of the font sizes in the images are too small and the clarity of the images is limited; updates are recommended.
This paper discusses the fusion of ESM and multi-scale operations. How does the computational complexity of the proposed algorithm compare with traditional detectors (such as the Canny edge detector) or deep learning-based methods? Is there a trade-off between computational efficiency and detection accuracy?
5.It is recommended that important results in all tables be highlighted in bold.
6.Although numerous references are cited in the paper, the coverage of key literature from the last three years is insufficient; it is recommended to supplement with recent research, particularly concerning the latest advancements in edge detection and noise handling.
7.It is advised to add a discussion section to further summarize and analyze the experimental results in depth.
8.In the conclusion section, it is recommended to further discuss the limitations of this paper and potential future work.

Author Response

请参阅附件
